# Divergent Influenza-Like Viruses of Amphibians and Fish Support an Ancient Evolutionary Association

**DOI:** 10.3390/v12091042

**Published:** 2020-09-18

**Authors:** Rhys Parry, Michelle Wille, Olivia M. H. Turnbull, Jemma L. Geoghegan, Edward C. Holmes

**Affiliations:** 1Australian Infectious Diseases Research Centre, School of Chemistry and Molecular Biosciences, The University of Queensland, Brisbane, QLD 4067, Australia; 2Marie Bashir Institute for Infectious Diseases and Biosecurity, School of Life & Environmental Sciences and School of Medical Sciences, The University of Sydney, Sydney, NSW 2006, Australia; 3Department of Biological Sciences, Macquarie University, Sydney, NSW 2109, Australia; olivia.turnbull@hdr.mq.edu.au; 4Department of Microbiology and Immunology, University of Otago, Dunedin 9016, New Zealand; jemma.geoghegan@otago.ac.nz; 5Institute of Environmental Science and Research, Wellington 5018, New Zealand

**Keywords:** *Orthomyxoviridae*, influenza, metatranscriptomics, fish, amphibians, evolution, phylogeny

## Abstract

Influenza viruses (family *Orthomyxoviridae*) infect a variety of vertebrates, including birds, humans, and other mammals. Recent metatranscriptomic studies have uncovered divergent influenza viruses in amphibians, fish and jawless vertebrates, suggesting that these viruses may be widely distributed. We sought to identify additional vertebrate influenza-like viruses through the analysis of publicly available RNA sequencing data. Accordingly, by data mining, we identified the complete coding segments of five divergent vertebrate influenza-like viruses. Three fell as sister lineages to influenza B virus: salamander influenza-like virus in Mexican walking fish (*Ambystoma mexicanum)* and plateau tiger salamander (*Ambystoma velasci*), Siamese algae-eater influenza-like virus in Siamese algae-eater fish (*Gyrinocheilus aymonieri*) and chum salmon influenza-like virus in chum salmon (*Oncorhynchus keta*). Similarly, we identified two influenza-like viruses of amphibians that fell as sister lineages to influenza D virus: cane toad influenza-like virus and the ornate chorus frog influenza-like virus, in the cane toad (*Rhinella marina)* and ornate chorus frog (*Microhyla fissipes)*, respectively. Despite their divergent phylogenetic positions, these viruses retained segment conservation and splicing consistent with transcriptional regulation in influenza B and influenza D viruses, and were detected in respiratory tissues. These data suggest that influenza viruses have been associated with vertebrates for their entire evolutionary history.

## 1. Introduction

Influenza viruses are segmented, negative-sense RNA viruses of the family *Orthomyxoviridae*, which also includes the genera *Quaranjavirus*, *Isavirus* and *Thogotovirus*, as well as a wide diversity of orthomyxo-like viruses found in invertebrates [1,2,3]. Influenza viruses are classified into four genera: *Alphainfluenzavirus*, *Betainfluenzavirus*, *Gammainfluenzavirus* and *Detainfluenzavirus* [4], each with a single ratified species: influenza A virus (IAV), influenza B virus (IBV), influenza C virus (ICV) and influenza D virus (IDV), respectively. All four influenza virus species have a conserved genomic arrangement, encompassing eight or seven segments for IAV/IBV and ICV, IDV, respectively. All influenza viruses contain segments polymerase basic protein 1 (PB1), polymerase basic protein 2 (PB2), and polymerase acidic protein (PA) (or P3 for ICV/IDV) encoding for the polymerase complex, NP (nucleoprotein), M (matrix) and NS (non-structural). This difference in segment number between IAV/IBV and ICV/IDV is due to differences in the glycoproteins. IAV and IBV contain both a hemagglutinin (HA) and neuraminidase (NA) segment, and the former contains the receptor binding domain and is central to virus entry and the latter for virus release. ICV and IDV have a single glycoprotein, hemagglutinin esterase fusion (HEF) which facilitates both virus entry and release, and as such have a genome comprising seven segments. Beyond the influenza viruses, more divergent orthomyxoviruses and orthomyxo-like viruses have other genomic modifications, such as Salmon isavirus (*Isavirus*), a pathogen of fish that encodes fusion (F) and a hemagglutinin esterase (HE), rather than HA, NA or HEF proteins [5]. Generally, influenza and influenza-like viruses encode proteins consistent with their phylogenetic position based on analyses of those that make up the RNA-dependent RNA polymerase (PB1). For example, Wuhan spiny eel influenza virus phylogenetically groups to a sister clade to IBV and all eight segments have been recovered [3].

Until recently, influenza viruses had been limited to a small group of pathogens with public health importance for humans, particularly those of pandemic and epidemic potential—influenza A virus (IAV, *Alphainfluenzavirus*) and influenza B virus (IBV, *Betainfluenzavirus*). Influenza A virus is an important, multi-host virus best described in humans, pigs, horses and birds [6,7,8,9]. These viruses are epidemic in humans and are closely monitored for pandemic potential. Influenza B viruses similarly cause yearly epidemics in humans [10], and their detection in both seals [11] and pigs [12] highlights the potential for non-human IBV reservoirs.

While IAV and IBV are subject to extensive research, far less is known about two additional influenza viruses: Influenza C virus (ICV, *Gammainfluenzavirus*) and Influenza D virus (IDV, *Deltainfluenzavirus*). Influenza C virus has been found in humans worldwide and is associated with mild clinical outcomes [13,14]. Based on its detection in pigs, dogs and camels, it is likely that this virus has a zoonotic origin [15,16,17,18,19]. Influenza D virus, first identified in 2015 [20], is currently thought to have a limited host range and is primarily associated with cattle and small ruminants [20,21,22], pigs [20], as well as dromedary camels [19]. Although there is no record of active infection in humans, there are indications of past infection based on serological studies [14,23,24].

Despite intensive research on the characterisation of influenza viruses, our understanding of their true diversity and their evolutionary history is undoubtedly limited. Of critical importance are recent metatranscriptomic (i.e., bulk RNA sequencing) studies that report the discovery of novel “influenza-like” viruses from both amphibian and fish hosts—Wuhan spiny eel influenza virus, Wuhan asiatic toad influenza virus and Wenling hagfish influenza virus [3]. Such a phylogenetic pattern, particularly the position of the hagfish (jawless vertebrate) influenza virus as the sister-group to the other vertebrate influenza viruses, is compatible with some virus–host co-divergence over millions of years. This, combined with frequent cross-species transmission, this suggests that a very large number of animal influenza-like viruses remain to be discovered.

Given the previous success of data-mining to identify a range of novel viruses, including rhabdoviruses [25], flaviviruses [26] and parvoviruses [27], we hypothesised we could mine publicly available transcriptome databases for evidence of undescribed vertebrate influenza viruses and that this would provide further insights into their evolutionary origins. Given the highly under-sampled nature of “lower” vertebrate hosts, such as fish and amphibians, we focused on these taxa. From this, we identified five novel influenza-like viruses that provide new insights into the evolution and host range of this important group of animal viruses.

## 2. Materials and Methods

### 2.1. Identification of Divergent Influenza-Like Viruses from Vertebrate De Novo Transcriptome Assemblies

To identify novel and potentially divergent vertebrate influenza viruses, we screened de novo transcriptome assemblies available at the National Center for Biotechnology Information (NCBI) Transcriptome Shotgun Assembly (TSA) Database (https://www.ncbi.nlm.nih.gov/genbank/tsa/) and the China National GeneBank (CNGB) Fish-T1K (Transcriptomes of 1000 Fishes) Project database (https://db.cngb.org) [28]. Amino acid sequences of the influenza virus reference strains—IAV A/Puerto Rico/8/34 (H1N1), IBV (B/Lee/1940), ICV (C/Ann Arbor/1/50) and IDV (D/bovine/France/2986/2012)—were queried against the assemblies using the translated Basic Local Alignment Search Tool (tBLASTn) algorithm under default scoring parameters and the BLOSUM45 matrix. For the TSA, we restricted the search to the Vertebrata (taxonomic identifier [taxid] 7742). Putative influenza-like virus contigs were subsequently queried using BLASTx against the non-redundant virus database. Two fish transcriptomes (BioProjects PRJNA329073 and PRJNA359138) were excluded because of the presence of reads that were near identical to IAV, strongly suggestive of contamination.

### 2.2. Recovery of Additional Influenza-Like Virus Segments Through de Novo Assembly and Coverage Statistics

Putative influenza-like viruses identified in the transcriptome data were queried against genus-wide RNA-Seq samples deposited in the Sequence Read Archive (SRA) using the BLASTn tool. Raw fastq files originating from transcriptome sequencing libraries were downloaded and imported to the Galaxy Australia web server (https://usegalaxy.org.au/, v19.05). Sequencing adapters were identified using FastQC, and reads were quality trimmed using Trimmomatic (Galaxy v0.36.4) under the following conditions: sliding window = 4 and average quality = 20 [29]. For the recovery of full-length transcripts corresponding to putative influenza-like viruses, clean reads were then de novo assembled using Trinity (Galaxy v2.9.1) [30]. Only sequencing runs from the same BioProject, and samples originating from the same geographic location and time were used for assembly. Viruses that appeared in multiple BioProjects or between different studies were assembled separately. For virus genome statistics, clean reads were re-mapped against virus segments using the Burrow-Wheeler Aligner (BWA-MEM Galaxy v0.7.17.1) under default conditions, and the resultant binary alignment file was analysed with the bedtools (v2.27.1) genome coverage tool [31]. For the production of the salamander influenza-virus infection heat map of *Ambystoma velasci* tissues, quantitation of abundance for each library was calculated using the proportion of total mapped salamander influenza-like reads for each library and heatmaps were produced using R v3.5.3 integrated in RStudio v1.1.463 and ggplot2.

### 2.3. Influenza Virus Genome Annotation

Viral open reading frames (ORFs) were predicted using ORFFinder (https://www.ncbi.nlm.nih.gov/orffinder/). To characterise functional domains, predicted protein sequences were subjected to a domain-based search using the Conserved Domain Database v3.16 (https://www.ncbi.nlm.nih.gov/Structure/cdd/cdd.shtml) and cross-referenced with the PFam database (v32.0) hosted at (http://pfam.xfam.org/). Transmembrane topology prediction using the TMHMM Server v2.0 (www.cbs.dtu.dk/services/TMHMM/). For the identification of potential splice sites in influenza-like viruses, we used alternative isoforms assembled using Trinity and then manually validated for donor and acceptor sites using the Alternative Splice Site Predictor (ASSP) (http://wangcomputing.com/assp/index.html) [32] for NS and M segments guided by experimentally validated positions in IAV/IBV/ICV (reviewed in [33]). Viral genome sequences were deposited in GenBank under the accession numbers: MT926372-MT926409

### 2.4. Phylogenetic Analysis

Multiple sequence alignments of predicted influenza virus protein sequences were performed using MAFFT-L-INS-i (v7.471) [34]. Ambiguously aligned regions were removed using TrimAl (v1.3) under the automated 1 method [35]. Individual protein alignments were then analysed to determine the best-fit model of amino acid substitution according to the Bayesian Information Criterion using ModelFinder [36] incorporated in IQ-TREE (v2.1.1) [37] and excluding the invariant sites parameter. For the PB1, PA, NP segments the Le-Gascuel (LG) model [38] with discrete gamma model with 4 rate categories (+Γ_4_) and empirical amino acid frequencies (+F) was selected as the most suitable. For the HA/HEF alignments, the Whelan and Goldman (WAG) model [39] with a discrete gamma model with 4 rate categories (+Γ_4_) was selected. For the NA alignment, the FLU + Γ_4_ model was identified [40], and PB2 and NS (LG + Γ_4_). For the M1/M2 alignment, LG + FreeRate model (+R3) [41] was selected. Maximum likelihood trees were then inferred using IQ-TREE (v2.1.1) with Ultrafast bootstrap approximation (*n* = 40,000) [42]. The resultant consensus tree from combined bootstrap trees was visualised using FigTree v1.4 (http://tree.bio.ed.ac.uk/software/figtree/).

## 3. Results

### 3.1. Discovery and Annotation of Novel Influenza-Like Viruses in Vertebrates

We initially screened 776 vertebrate de novo assembled transcriptomes deposited in the Transcriptome Shotgun Assembly Sequence Database (TSA) and a further 158 transcriptomes available on the Fish-T1K Project. Within the TSA entries, the most abundant assemblies were from the bony fish (Actinopterygii), accounting for 346 transcriptomes. We also screened the amphibian (*n* = 57) and cartilaginous fish (*n* = 9) transcriptomes. From these libraries, we detected influenza-like viruses in 0.6% of the bony fish libraries (*n* = 2) and in 7% (*n* = 4) of the amphibian libraries.

Our transcriptome mining identified fragments of divergent IBV-like PB1 and PB2 segments from an unpublished transcriptome of ear and neuromast samples from an *Ambystoma mexicanum* (Mexican walking fish or Axolotl) laboratory colony housed by The University of Iowa (BioProject Accession: PRJNA480225). BLASTx analysis of these IBV-like fragments revealed 76.32% amino acid identity to the PB1 gene of IBV (top hit GenBank accession ANW79211.1; *E*-value: 0; query cover 100%) and 60.94% to the IBV PB2 (top hit GenBank accession: AGX15674.1; *E*-value: 2 × 10^−49^; query cover: 83%) (Table 1). To recover additional segments and full-length PB1 and PB2 genes, we used BLASTn to screen 2037 *Ambystoma* RNA-Seq libraries from several *Ambystoma* species available on the SRA. Accordingly, we were able to identify this tentatively named salamander influenza-like virus in 139/2037 (6.8%) of screened libraries. BLASTn positive libraries were subsequently de novo assembled, allowing the recovery of all eight segments of this virus (Figure 1; Table 1). In addition to *A. mexicanum* samples infected with salamander influenza-like virus, we identified a smaller number of reads from the Anderson’s salamander (*Ambystoma andersoni*) and spotted salamander (*Ambystoma maculatum*) libraries [43]. We identified a variant of salamander influenza virus from RNA-Seq data generated from laboratory colonies of another neotenic salamander, plateau tiger salamander (*Ambystoma velasci*) [44] with nucleotide sequence identity between both variants of all segments between 89 and 94% at the nucleotide level (PB1 90%, PB2 91%, PA 90%, NP 91%, NA 90%, HA 89%, NS 94%, M 93%). The presence of this viral genome in up to four species of *Ambystoma* colonies indicates that it potentially has a wide host range across this host genus. Additionally, as the majority of *A. mexicanum* colonies originate from the *Ambystoma* Genetic Stock Center (AGSC, University of Kentucky, KY, USA) [45,46,47,48] and in some cases co-housed together with other species positive for this virus [43], this virus may circulate between numerous laboratory-reared salamander colonies.

We similarly identified an influenza virus in the Siamese algae-eater (*Gyrinocheilus aymonieri*) (SRA accession: SRR5997773), tentatively termed Siamese algae-eater influenza-like virus, sampled as part of the Fish-T1K Project (Table 1) [28]. While there are limited meta-data available, the library was prepared from gill tissues. There was no evidence of this virus in any other Siamese algae-eater samples available on the SRA (*n* = 2) through BLASTn analysis.

The third IBV-like virus was identified in chum salmon (*Oncorhynchus keta)*, provisionally termed chum salmon influenza-like virus (Table 1). While the geographic origin of the samples is unknown, the meta-data from this library indicate that the RNA-Seq originates from gill tissues of alevin stage chum salmon, transferred to the Korea Fisheries Resources Agency (FIRA) laboratory and reared in tanks with recirculating freshwater [49]. We identified chum salmon influenza-like virus sequences in one of the three libraries from this project (SRA Accession: SRR6998471). Coding-complete sequences of all segments were de novo assembled with average coverage ranging between ~25× in the PB1 fragment and 931× in the M segment (Appendix A). Re-mapping clean reads to the putative virus genome indicated that chum salmon influenza-like virus RNA corresponded to 0.045% of the library (45,151/100,244,990 mapped reads).

In addition to novel viruses related to IBV, we identified two influenza viruses of amphibians that exhibited relatively high pairwise amino acid identity to segments from IDV and ICV (Table 1). This result is particularly noteworthy as the previously identified amphibian influenza virus, Wuhan asiatic toad influenza virus, was more closely related to IBV [3]. Cane toad influenza-like virus was identified in four larval samples of cane toad (*Rhinella marina*), from two geographic locations in Australia: Innisfail, Queensland, Australia (SRA accessions: SRR5446725, SRR5446726) and Oombulgurri, Western Australia, Australia (SRA accessions: SRR5446727, SRR5446728) [50]. We did not identify any virus reads in any adult tissue samples taken in Australia nor from libraries constructed from cane toad samples from Macapa city, Brazil, from the same study [50]. Similarly, we did not recover any cane toad influenza-like virus reads in any of the 16 liver tissue samples from Australia reported in another metatranscriptomic study, which aimed to reveal the virome of the cane toad [53].

The other IDV-related virus identified was tentatively named ornate chorus frog influenza-like virus, detected in the ornate chorus frog (*Microhyla fissipes*) (Table 1). This virus was identified in two sequencing projects, both of which were from samples of ornate chorus frog collected from paddy fields in Chengdu, China, albeit at different times; May 2014 (BioProject accession: PRJNA295354) [51] and June 2016 (BioProject accession: PRJNA386601) [52]. Between both data sets, the largest number of ornate chorus frog influenza-like virus reads originated from the lung tissue of the stage 28 tadpole (SRA accession: SRR5557878) in the second study (June 2016 [52]), representing 0.13% of all reads (81,860/61,952,548). While there were fewer reads mapping to the ornate chorus frog influenza-like virus in whole tissues of three developmental stages in the study carried out in 2014 [51], in whole organism samples, the highest viral abundance was in the metamorphic climax (1462/59,993,514 reads) (SRA accession: SRR2418623), and pre metamorphosis developmental phases (1113/63,073,925 reads) (SRA accession: SRR2418554), with only seven reads identified in the complete metamorphosis library (7/56,000,000 reads) (SRA accession: SRR2418812). The nucleotide identity of ornate chorus frog influenza-like virus variants between the 2014 [51] and June 2016 [52] data sets was between 93.36 and 97.10% for all segments (PB1 96.98% PB2 93.95% P3 93.36% NP 93.83% HEF 91.87% NS 97.10% M 96.29%).

### 3.2. The Genome Organisation and Transcription of Novel Influenza-Like Virus Genes Are Highly Conserved

Through a combination of de novo assembly and tBLASTn analysis, we were able to identify and assemble the complete coding segments of all novel influenza-like genomes described in this study (Figure 1, Appendix A, Appendix A). For the salamander influenza-like virus, Siamese algae eater influenza-like virus and chum salmon influenza-like viruses, all eight segments with genome arrangements similar to that of IAV and IBV were identified (Figure 1). Prediction of the ORFs from all segments and protein domain analyses suggested homology between these putative virus proteins (Appendix A, Appendix A). Additionally, we identified the glycoprotein NB, which is encoded through a polycistronic mRNA of the NA segment of IBVs (Figure 1). The M2 domain of chum salmon influenza-like virus was not identified through BLASTp, likely due to high levels of sequence divergence. However, we did identify the N terminus using a domain search (Appendix A). In contrast, we identified seven segments each for cane toad influenza-like virus and ornate chorus frog influenza-like virus, which are characteristic of ICV/IDV encoding seven segments (Figure 1). We recovered all ORFs expected of viruses similar to ICV and IDV (Appendix A, Appendix A).

There are a number of important differences in the composition of segments among IAV/IBV and ICV/IDV. For example, segment 3 is differentiated into “PA” or “P3” by isoelectric points. The isoelectric points of ornate chorus frog influenza-like virus (~6.3) and cane toad influenza-like virus (~6) are similar to those of the P3 segment of ICV/IDV, while those of salamander influenza-like virus (~5.6), Siamese algae eater influenza-like virus (~5.3) and chum salmon influenza-like viruses (~5.4) had isoelectric points more consistent with PA of IAV (~5.4) and IBV (~5.5). We were able to assemble two discrete isoforms of the *NS* gene for all viruses and manually validate the splice junctions as bona fide through the identification of splicing donor and acceptor sites (Appendix A), with the exception of chum salmon influenza-like virus, where we could only predict putative donor and acceptor isoform locations on the NS segment (Figure 1). The M segment in ICV/IDV-related viruses is also post-transcriptionally spliced to encode the M1 protein, and we were able to assemble two isoforms of the M segment from cane toad influenza-like virus and ornate chorus frog influenza-like virus.

### 3.3. Phylogenetic Analysis of Novel Influenza-Like Viruses Suggests a History of Genomic Reassortment

One of the most striking observations of our study was that three novel influenza viruses from amphibians and fish fell basal to IBV in the phylogenetic analysis, although none of the relevant bootstrap values were exceptionally high (i.e., <75%) (Figure 2), and with pairwise amino acid sequences similarities to the PB1 of IBV ranging from 67 to 75%. The placement of these viruses, in addition to Wuhan spiny eel influenza virus, at the base of the IBV lineage, strongly suggests that these viruses have a long evolutionary history in vertebrates, although with frequent host-jumping. Indeed, it is notable that salamander influenza-like virus is the now the closest animal relative to IBV. Similarly, two other influenza-like viruses from amphibians were basal to IDV, this time with strong (>90%) bootstrap support, and exhibited 77–80% amino acid similarity in the PB1 segment.

These novel influenza viruses largely occupy consistent phylogenetic placements across segments (Figure 3). However, chum salmon influenza-like virus is the sister-group to IBV in the polymerase (PB2, PB1, PA) and NA segments, but is the sister-group to both IAV/IBV in the NP segment. Additionally of note was that the two amphibian influenza-like genomes are the sister lineages to IDV in all segments with the exception of the HEF, in which they are the sister lineages to the clade containing ICV/IDV.

### 3.4. Multi-Tissue RNA-Sequencing of Developmental Plateau Tiger Salamander Libraries Suggests Conserved Influenza Virus Tropism in Vertebrates

To gain further insight into the tissue tropism of these divergent influenza viruses, we screened libraries from a multi-tissue and developmental library of the plateau tiger salamander. This dataset comprised 86 individual samples from gills, hearts and lungs at different stages of pre metamorphosis, pro metamorphosis, metamorphosis and late metamorphosis [44]. We identified RNA originating from salamander influenza-like virus abundantly in the libraries originating from gills at the late metamorphosis stage with an average number of the reads in all six libraries corresponding to approximately 0.22% of all reads (Figure 4). The second most abundant tissue and developmental stage was post metamorphosis stage lung tissue which, on average, had 0.013% of all reads correspond to salamander influenza-like virus. The only other tissue and developmental stage with all segments represented were the gills from the pre metamorphosis stage with an average ~0.003% of all reads originating from salamander influenza-like virus.

## 4. Discussion

We identified five novel influenza viruses in fish and amphibians through mining publicly available meta-transcriptomic data. Despite intensive research on influenza viruses for almost a century, it is only recently that these viruses have been identified in hosts other than birds and mammals. In particular, the recent identification of divergent influenza-like viruses in fish and amphibians [3], as well as even more divergent orthomyxoviruses in invertebrates [2], suggests that divergent members of this virus family may infect a wide range of animal hosts.

These data provide valuable insights into the evolution of influenza viruses across the vertebrates. First, the fact that the phylogeny of the IBV-like viruses in part follows the phylogeny of the hosts from which they are sampled, with the salamander influenza-like viruses falling as the sister-group to the mammalian viruses and the fish viruses occupying basal positions in that group as a whole, is compatible with a process of virus–host co-divergence that likely extends hundreds of millions of years, albeit with frequent host-jumping. Under these circumstances, it must also be the case that there are many more vertebrates that carry IBV-like viruses that have yet to be investigated. Similarly, we identified two amphibian viruses that fall as sister lineages to IDV, again suggestive of ancient virus-host associations and highly limited sampling to date. Whether the same pattern will also be true of the influenza A virus group is unclear and will only be resolved with additional sampling. It is notable that the Wenling hagfish virus, from a jawless vertebrate (a hagfish), occupies the basal position in all of the vertebrate influenza virus, again suggesting that these viruses have been in existence for hundreds of millions of years.

Despite detecting divergent influenza viruses, the stability and conservation of genome segments and transcriptional regulation of viral genes through splicing is conserved. For example, salamander influenza-like virus, Siamese algae-eater influenza-like virus and chum salmon influenza-like virus, which all fall basal to IBV, possess eight segments encoding all the predicted/required ORFs including NB, which is a glycoprotein encoded on the *NA* gene of IBV only [54]. Similarly, cane toad influenza and ornate chorus frog influenza had only seven segments and all ORFs were consistent with ICV and IDV. Overall, despite finding these viruses in non-mammalian hosts, the viral structure is highly consistent with other influenza viruses, strongly supporting the inclusion into these viral genera.

These new data also raise questions of whether influenza and influenza-like viruses have the same tissue tropism among all vertebrate hosts. We detected all novel influenza viruses in respiratory tissues (i.e., gills and lungs) of their hosts: the only exception was the cane toad influenza-like virus, for which we do not have tissue-associated metadata. The most robust support for the conservation of influenza-like virus infection in respiratory tissues was in the analysis of the multi-tissue transcriptome of the plateau tiger salamander [44]. Before and during metamorphosis, salamanders rely on gills for respiration. During metamorphosis, there is a significant reduction in gill length, and in post-metamorphosis, they rely on lungs for respiration [44,55]. Based on the expectation that we would find these viruses in respiratory tissues, we found salamander influenza-like virus in the gills of metamorphosing salamanders and the lungs following metamorphosis. This suggests that all influenza viruses primarily infect the respiratory tissues, consistent with other influenza viruses [56]. To date, the only exception are IAV infections, which infect the surface epithelium of the gastrointestinal tract in wild birds [57], and gut-associated lymphoid tissue and the squamous epithelium of the palatine tonsils in bats [58]. That IAV has tropism for gastrointestinal tissues in the natural wild bird reservoir seems to be an outlier compared to other influenza viruses. Indeed, IAV infections in all other hosts (humans, pigs, horses, dogs, and even poultry) are respiratory.

Finally, it is interesting to speculate on whether these influenza-like viruses are host specialist or generalist viruses. Influenza A–D viruses have extensive host ranges, including an array of mammalian species, particularly those associated with food production (i.e., cattle and swine), and in the case of IAV, wild and food production associated birds (i.e., poultry) [6,7,8,9,20,21,22]. Salamander influenza-like virus also appears to be a multi-host virus, being found in four salamander species raised in laboratory colonies, with the greatest abundance in the Mexican walking fish and plateau tiger salamander. While these species have very different life-history strategies, and their ranges do not overlap in nature [59,60], both are abundant in the pet trade and used in laboratory research. Thus, breeding facilities may be an important source of these viruses. Whether this multi-host trait is correlated with the use of breeding facilities in salamanders, a parallel for animal production systems and influenza A-D viruses is unclear. Further research is certainly warranted to determine whether a broad host range is a feature of all influenza viruses, including those found in “lower” vertebrates.

## 5. Conclusions

By mining transcriptome data, we identified five novel influenza-like viruses in fish and amphibians. There is clearly scope and need for more dedicated work on the viruses revealed here, including more dedicated studies to demonstrate virus function, disease association, prevalence and host range. The data generated in this study strongly suggest that influenza-like viruses can infect diverse classes of vertebrates and that influenza viruses have been associated with vertebrates for perhaps their entire evolutionary history. While the public health and economic ramifications are undoubtedly more significant for mammalian influenza viruses, only by looking at other vertebrate classes and other animal lineages will we fully understand the origins and evolution of this hugely important group of viruses.

## Figures and Tables

**Figure 1 viruses-12-01042-f001:**
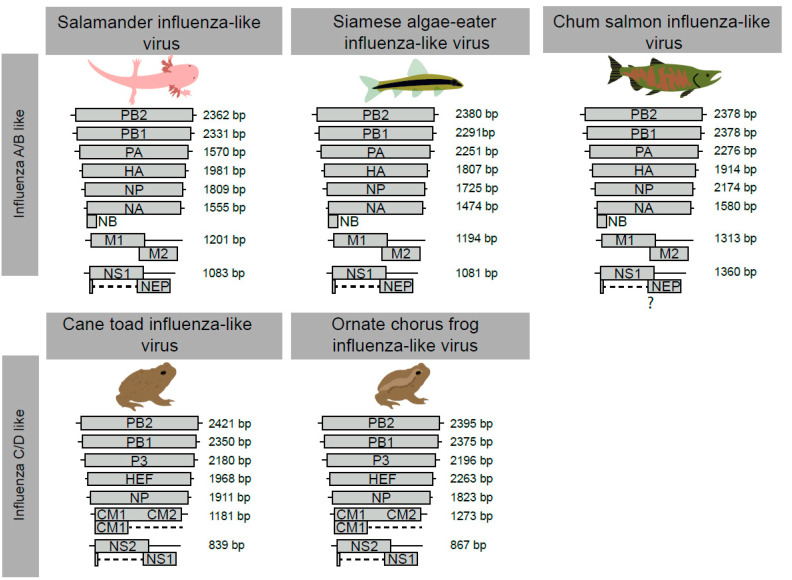
Genome architecture of the novel influenza-like viruses identified here. Genome architecture of influenza A virus (IAV)/influenza B virus (IBV) and influenza C virus (ICV)/influenza D virus (IDV) is shown on the left. For each novel virus, we provide the segment name and size. PB1, RNA-dependent RNA polymerase basic subunit 1; PB2, RNA-dependent RNA polymerase basic subunit 2; PA, RNA-dependent RNA polymerase acidic subunit; P3, Polymerase protein 3; NP, nucleoprotein; HA, hemagglutinin; NB, glycoprotein NB; HEF, hemagglutinin esterase; NA, neuraminidase; M, matrix; CM1, viral matrix protein; NS, non-structural protein; NEP nuclear export protein. Detailed annotations for each virus are presented in Appendix A.

**Figure 2 viruses-12-01042-f002:**
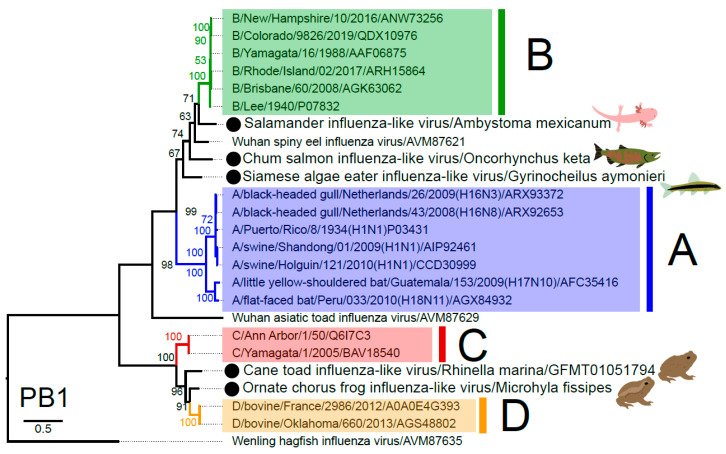
Phylogenetic relationships of vertebrate influenza-like viruses. Maximum likelihood tree of the PB1 segment, which encodes the RNA-dependent RNA polymerase, of various influenza-like viruses. Lineages corresponding to IAV, IBV, ICV, IDV are coloured blue, green, red and orange, respectively. Viruses identified in this study are denoted by a black circle and pictogram of the host species of the library. Wenling hagfish influenza virus is set as the outgroup as per Shi et al. [3]. The scale bar indicates the number of amino acid substitutions per site.

**Figure 3 viruses-12-01042-f003:**
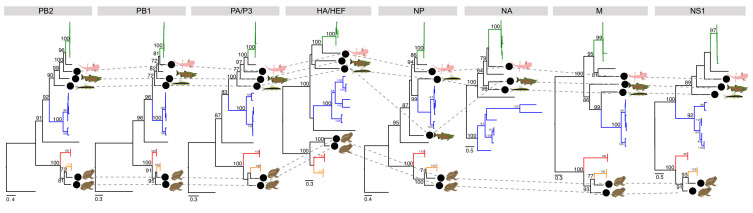
Phylogenetic trees for segments in cases where complete coding genomes could be recovered. Maximum likelihood trees for each segment ordered by (segment) size. Lineages are coloured by influenza virus, in which IAV, IBV, ICV, IDV are coloured blue, green, red and orange, respectively. Viruses identified in this study are denoted by a black circle and pictogram of the host species of the library. The phylogenetic position of each virus is traced across the trees with grey dashed lines. ICV, IDV, cane toad and chorus frog influenza-like viruses do not have an NA segment. The scale bar for each tree indicates the number of amino acid substitutions per site. Where possible, the trees are rooted using the hagfish influenza virus (PB2, PB1, PA/P3, NP). The HA/HEF, M and NS trees were midpoint rooted.

**Figure 4 viruses-12-01042-f004:**
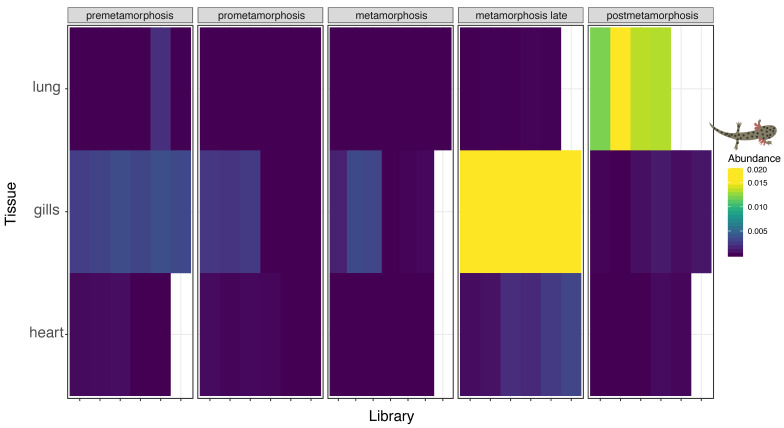
Heat map of the abundance of salamander influenza-like virus reads in the multi-tissue transcriptome library of developmental stages of plateau tiger salamander [44]. Read abundance for this virus was highest in the gills of late metamorphosis salamanders, and lungs of post metamorphosis salamander, strongly suggesting tropism for respiratory epithelium.

**Table 1 viruses-12-01042-t001:** Metadata for the novel influenza viruses identified in this study.

Virus Name	Host	Tissue Sampled	Reference	BLASTp Output of Predicted Amino Acid Identity of Polymerase Genes			
				Segment	GenBank Accession	Coverage (%); Identity (%)	*E*-value
Salamander influenza-like virus	Mexican walking fish (*Ambystoma mexicanum*)	Various	[43,45,46,47,48]	PB1 Influenza B virus (B/California/24/2016) PB2 Influenza B virus (B/Sydney/19/2011) PA Influenza B virus (B/Indiana/07/2016)	QHI05420 AZY32600 ANW74127	100%; 75.70% 99%; 61.83% 97%; 59.64%	0.0 0.0 0.0
	Plateau tiger salamander (*Ambystoma velasci*)	Various	[44]				
Siamese algae-eater influenza-like virus	Siamese algae-eater (*Gyrinocheilus aymonieri*)	Gills	[28]	PB1 Influenza B virus (B/California/24/2016) PB2 Influenza B virus (B/New York/1121/2007) PA Wuhan spiny eel influenza virus	ANW79211 AHL92298 AVM87622	99%; 69.46% 99%; 48.58% 96%; 53.89%	0.0 0.0 0.0
Chum salmon influenza-like virus	Chum salmon (*Oncorhynchus keta*)	Gills	[49]	PB1 Influenza B virus (B/Iowa/14/2017) PB2 Influenza B virus (B/Memphis/5/93) PA Influenza B virus (B/Taiwan/45/2007)	QHI05420 AAU94860 ACO06009	100%; 69.50% 99%; 57.05% 98%; 51.88%	0.0 0.0 0.0
Cane toad influenza-like virus	Cane Toad (*Rhinella marina*)	Tissue unknown, Larval	[50]	PB1 Influenza D virus (bovine/Mexico/S7/2015) PB2 Influenza D virus (bovine/Kansas/14-22/2012) P3 Influenza D virus (bovine/Yamagata/10710/2016)	AMN87903 AIO11621 BBC14929	99%; 77.03% 99%; 62.87% 100%; 62.66%	0.0 0.0 0.0
Ornate chorus frog influenza-like virus	Ornate chorus frog (*Microhyla fissipes*)	Larval, Lung	[51,52]	PB1 Influenza D virus (bovine/Mississippi/C00046N/2014) PB2 Influenza D virus (swine/Italy/173287-4/2016) P3 Influenza D virus (D/bovine/Shandong/Y217/2014)	ALE66333 AON76692 AIE52099	98%; 81.91% 99%; 68.22% 99%; 63.78%	0.0 0.0 0.0

PB1, polymerase basic protein 1; PB2, polymerase basic protein 2; PA, polymerase acidic protein; P3, polymerase protein 3.

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
