# Peer review of "Divergent Influenza-Like Viruses of Amphibians and Fish Support an Ancient Evolutionary Association"

_viruses, 2020, doi:10.3390/v12091042_

Round 1
Reviewer 1 Report
In the submitted manuscript, Parry et al. identified influenza viruses in fish and amphibians. Further, the viruses are phylogenetically distinct from previously known viruses. Although the study seems sound and their findings contribute significantly to the field, I have several concerns to be addressed for the publication.
Major points:
(Lines 33-34 and 293-304) The authors suggest host-virus co-divergence occurred based on the topology of phylogenetic trees they constructed. They insist that the co-divergence happened for IBV and IDV independently. If that is true, this means that divergence of IBV and IDV had occurred before the divergence among fish, amphibians, reptiles, and mammals. That does not seem likely to me. Also, high identity of genetic sequences and well-conserved genomic organizations rather suggest recent (still, may be >1000 years, but not millions of years) introduction of the virus to new hosts. Of course, we don’t know the right answer. The authors should at least discuss other possibilities than co-divergence.
(Introduction) The authors should explain a natural host of IAV, genome organizations of influenza viruses, and current hypothesis of evolutionary history of the virus as background in the Introduction section.
(Lines 143-158, 183-198, 202-204, Figure 1) From the authors’ descriptions, some viruses were found in different samples from different individuals, at least for salamander virus and frog virus. Then, were the assembled genome of the viruses made using multiple different samples? The author should clarify the point. And, if so, the point should be mentioned as a limitation in the Discussion section.
(Lines 220-223) “With the exception of chum salmon influenza-like virus, we were able to assemble two discrete isoforms of the NS gene in all our assembles and manually validate the splice junctions…”
Then, how did the author determine there were NS1 and NEP for the NS segment of the salmon virus as shown in Figure 1?
Minor points:
(Lines 316-318, 326-328) “As with other influenza viruses…” “To date, the only exception are…” Today, wild aquatic birds are considered to be a natural host of IAV and source of its diversity. It sounds strange to me to call their histological tropism “exception.”
(Results) There are too many “finally” and “final” in the section.
(Lines 152-153) Information about identity among variants of salamander virus is not described in the Table 1.
Author Response
REVIEWER 1
In the submitted manuscript, Parry et al. identified influenza viruses in fish and amphibians. Further, the viruses are phylogenetically distinct from previously known viruses. Although the study seems sound and their findings contribute significantly to the field, I have several concerns to be addressed for the publication.
Major points:
(Lines 33-34 and 293-304) The authors suggest host-virus co-divergence occurred based on the topology of phylogenetic trees they constructed. They insist that the co-divergence happened for IBV and IDV independently. If that is true, this means that divergence of IBV and IDV had occurred before the divergence among fish, amphibians, reptiles, and mammals. That does not seem likely to me. Also, high identity of genetic sequences and well-conserved genomic organisations rather suggest recent (still, may be >1000 years, but not millions of years) introduction of the virus to new hosts. Of course, we don’t know the right answer. The authors should at least discuss other possibilities than co-divergence.
Response: We must respectfully disagree with the reviewer on this point. First, we do not ‘insist’ on co-divergence. Rather, we make it clear throughout the paper that the long-term evolution of influenza virus reflects a combination of both co-divergence and cross-species transmission (i.e. host-jumping). For example, in lines 76-80 we state: “Such a phylogenetic pattern, particularly the position of the hagfish (jawless vertebrate) influenza virus as the sister-group to the other vertebrate influenza viruses, is compatible with virus-host co-divergence over millions of years. This, combined with relatively frequent cross-species transmission, suggests that a very large number of animal influenza-like viruses remain to be discovered”. In the revised version of the paper we have deleted the word “relatively”. Similarly, in lines 316-317 we say: “This is compatible with a process of virus-host co-divergence that likely extends hundreds of millions of years, albeit with relatively frequent host-jumping”. There is nothing in the data we present that makes us change this view, which we strongly believe explains the patterns observed in IBV and IBV. In addition, the reviewer forgets that the sister-group to these vertebrate viruses are a set of viruses from invertebrates, exactly as expected under co-divergence over hundreds of millions of years.
We also strongly contend that the idea that these viruses have diversified on time scales of 1000s of years is untenable given their extensive sequence divergence and the wide range of hosts infected. More directly, evolutionary rates are strongly time-dependent in RNA viruses such that we progressively and hugely underestimate true divergence times. This point is clearly explained here: https://royalsocietypublishing.org/doi/full/10.1098/rspb.2014.0732
However, we have clarified the relevant statements, making it clear throughout that the diversity of influenza viruses is likely due to a combination of both co-divergence and cross-species transmission. Indeed, we plainly state that cross-species transmission is a common occurrence in these viruses and in part explains the diversity observed (Lines 79-81, lines 260-261, lines 318-319).
(Introduction) The authors should explain a natural host of IAV, genome organizations of influenza viruses, and current hypothesis of evolutionary history of the virus as background in the Introduction section.
Response: Lines 60-61 discuss the natural hosts of influenza A. Lines 73-81 currently explain the evolutionary history of influenza viruses. As requested, we have added a short section to the Introduction to explain the difference genome organisations of influenza viruses (lines 43-56)
(Lines 143-158, 183-198, 202-204, Figure 1) From the authors’ descriptions, some viruses were found in different samples from different individuals, at least for salamander virus and frog virus. Then, were the assembled genome of the viruses made using multiple different samples? The author should clarify the point. And, if so, the point should be mentioned as a limitation in the Discussion section.
Response: To ensure we did not produce chimeric viruses, we only pooled sequencing runs from the same BioProject and also from the same geographic location and time (See Supplementary Figures). For example, for Cane toad influenza-like virus we only used sequencing data from whole larval samples from Innisfail, QLD. Two different geographic locations are available from this data set. To make this clearer, we have added Lines 111-113 to the relevant section in the methods, which states: “Only sequencing runs from the same BioProject, and samples originating from the same geographic location and time were used for assembly. Viruses that appeared in multiple BioProjects or between different studies were assembled separately”
(Lines 220-223) “With the exception of chum salmon influenza-like virus, we were able to assemble two discrete isoforms of the NS gene in all our assembles and manually validate the splice junctions…” Then, how did the author determine there were NS1 and NEP for the NS segment of the salmon virus as shown in Figure 1?
Response: For Chum salmon influenza virus, we used isoform donor and acceptor prediction tools as outlined in the Methods as in silico prediction. For all the other influenza-like viruses there was harmony between the assembled transcripts and also predicted splice and acceptor sites using this tool, so it was thought that this was a reasonable prediction. We agree that this may not be a confident annotation we have now changed the wording to indicate that this is only predicted: “with the exception of chum salmon influenza-like virus, where we could only predict putative donor and acceptor isoform locations on the NS segment” (Lines 242-243)
Minor points:
(Lines 316-318, 326-328) “As with other influenza viruses…” “To date, the only exception are…” Today, wild aquatic birds are considered to be a natural host of IAV and source of its diversity. It sounds strange to me to call their histological tropism “exception.”
Response: The reviewer is correct that wild birds are believed to be the natural host of IAV and source of its genetic diversity. However, if we are to look across all influenza viruses, the replication of IAV in the gastrointestinal tract of wild birds seems to be the exception compared to the strong respiratory tropism seen in influenza B, C, D viruses. Further, despite wild birds being the natural reservoir of IAV, there is strong tropism for respiratory tissues in all other hosts (humans, pigs, horses, dogs, and even poultry). We have added the statement: “That IAV has tropism for gastrointestinal tissues in the natural wild bird reservoir seems to be an outlier compared to other influenza viruses. Indeed, IAV infections in all other hosts (humans, pigs, horses, dogs, and even poultry) are respiratory.” to the end of this paragraph (lines 349-351)
(Results) There are too many “finally” and “final” in the section.
Response: Fair point. We have removed the majority of occurrences of this expression.
(Lines 152-153) Information about identity among variants of salamander virus is not described in the Table 1.
Response: We have removed the reference to Table 1
Reviewer 2 Report
Parry and Wille et al. have written a well presented and robust manuscript detailing their findings from data mining of RNA sequencing data, data that was presumably originally generated for reasons other than discovery of novel influenza-like viruses. They add to the growing appreciation of the role of animals, other than mammals or birds, to influenza virus evolution. It is interesting to see that even these novel influenza-like viruses 'retain' tropism for respiratory tissues. Would the authors consider adding some context to the generation of RNA seq data in the introduction i.e. why do groups/initiatives generate RNA seq data in the first place, and maybe give an appreciation to this resource in the discussion? Do the authors think it is worth speculating on the potential of actively sequencing influenza-like viruses in these now known reservoirs with bespoke primers? Perhaps to see how prevalent e.g. influenza B viruses are in Mexican walking fish?
Author Response
REVIEWER 2
Parry and Wille et al. have written a well presented and robust manuscript detailing their findings from data mining of RNA sequencing data, data that was presumably originally generated for reasons other than discovery of novel influenza-like viruses. They add to the growing appreciation of the role of animals, other than mammals or birds, to influenza virus evolution. It is interesting to see that even these novel influenza-like viruses 'retain' tropism for respiratory tissues.
Would the authors consider adding some context to the generation of RNA seq data in the introduction i.e. why do groups/initiatives generate RNA seq data in the first place, and maybe give an appreciation to this resource in the discussion?
Response: We discuss the original purpose of many of the libraries mined here in detail in the Results section. RNA-Seq has enormous utility and is a standard method in the toolkit of biologists. Accordingly, a discussion of the original purpose of studies undertaken by other researchers does not seem central to the message of this article.
Do the authors think it is worth speculating on the potential of actively sequencing influenza-like viruses in these now known reservoirs with bespoke primers? Perhaps to see how prevalent e.g. influenza B viruses are in Mexican walking fish?
Response: We would certainly encourage additional work on influenza viruses in fishes and amphibians, including more dedicate work into virus function, disease association, prevalence and ecology. With enhanced methods for virus detection from metagenomic data, virus discovery will undoubtedly outstrip the pace of these more detailed studies. We have added relevant a statement to the conclusion: “There is clearly scope and need and for more dedicated work on the viruses revealed here, including more dedicated studies to demonstrate virus function, disease association, prevalence and host range.” (lines 366-368)
Reviewer 3 Report
The manuscript by Parry et al. describes the detection of novel influenza-like viruses of amphibians and fish from mining publicly available metatranscriptomic data. Their findings are exciting, and contribute to fill in the gaps of the evolutionary history of influenza viruses. Their findings are also coherent with other recent reports of influenza-like viruses in eel and amphibians. Overall the manuscript is well written, and the methodology and results are well explained. One general comment:
For the viruses that were recovered from multiple SRA samples, I'm curious if the authors detected some segments in more than one sample, and whether these were divergent (or identical) within a certain genus (i.e. Ambystoma). Perhaps assembly of complete segments was only possible fom the combined samples, but this could shed some light on the origin and diversity of these viruses within a particular host (i.e. salamander). Also are these only "new world" salamanders? the authors suggest that the viruses might circulate among numerous laboratory-reared colonies. Perhaps looking at the host range of these species in nature may reveal something about their potential origin. Lastly, although I'm not suggesting this to be part of this manuscript, it would be interesting to demonstrate infection in these colonies, and validate the findings of the current paper.
Specific comments to the text:
Line 153: Perhaps the authors should be more careful with the language "The presence of infection" as "infection" as such has not been demonstrated. Consider using the "presence of virus RNA".
Line 170 "...all segmnets were convincing de novo assembled..." could you detail the criteria for a convincing assembly? A coverage threshold per base, etc. These parameters could serve as a reference for other similar studies.
Figure 2. Wenling hagfish influenza virus is used as an outgroup for their phylogenies. I'm curious what the level of divergence is of these novel viruses with the Isaviruses? for example how much more we have to look to reconstruct the relationships and evolution of the Orthomyxoviridae as a whole. Depending on the level of divergence among these, the authors might want to include or not a related statement.
Figure 3. Please increase the size of the scale bars and support values. They are not readable in the current version.
Author Response
REVIEWER 3
The manuscript by Parry et al. describes the detection of novel influenza-like viruses of amphibians and fish from mining publicly available metatranscriptomic data. Their findings are exciting, and contribute to fill in the gaps of the evolutionary history of influenza viruses. Their findings are also coherent with other recent reports of influenza-like viruses in eel and amphibians. Overall the manuscript is well written, and the methodology and results are well explained. One general comment:
For the viruses that were recovered from multiple SRA samples, I'm curious if the authors detected some segments in more than one sample, and whether these were divergent (or identical) within a certain genus (i.e. Ambystoma). Perhaps assembly of complete segments was only possible from the combined samples, but this could shed some light on the origin and diversity of these viruses within a particular host (i.e. salamander).
Response: We appreciate the concern about the pooling of samples which may produce chimeric virus assemblies. For all virus genomes, we only pooled sequencing runs from the same BioProject, originating from the same geographic location and sampling time. We did not need to pool across BioProjects or between different datasets from different studies as we had enough depth of sequencing (Supplementary Figures 1-5). The details of the exact libraries used for assembly, the exact metadata and paper origins as well as metrics such as genome coverage are outlined in the Supplementary Figures. To make this clearer, we have added Lines 111-113 to the relevant section in the methods.
Also are these only "new world" salamanders? the authors suggest that the viruses might circulate among numerous laboratory-reared colonies. Perhaps looking at the host range of these species in nature may reveal something about their potential origin.
Response: Our study was limited to mining publicly available data. We believe that there is enormous potential for more work on these viruses, including large screening studies of salamanders. While we encourage these efforts, they are sadly outside the scope of our study.
Lastly, although I'm not suggesting this to be part of this manuscript, it would be interesting to demonstrate infection in these colonies, and validate the findings of the current paper.
Response: The reviewer raises an excellent point – the limitation of mining publicly available data is that we may reveal a sequence but cannot demonstrate infection (which clearly require virological tools). This work is certainly warranted, but outside the scope of our study. We have therefore added a sentence to the conclusion to indicate these suggestions: “There is clearly scope and need and for more dedicated work on the viruses revealed here, including more dedicated studies to demonstrate virus function, disease association, prevalence and host range” (lines 366-368)
Specific comments to the text:
Line 153: Perhaps the authors should be more careful with the language "The presence of infection" as "infection" as such has not been demonstrated. Consider using the "presence of virus RNA".
Response: Good point. We have replaced “The presence of infection” with “The presence of this viral genome”
Line 170 "...all segmnets were convincing de novo assembled..." could you detail the criteria for a convincing assembly? A coverage threshold per base, etc. These parameters could serve as a reference for other similar studies.
Response: To remove ambiguity with what constitutes “convincing,” we have removed this from the manuscript and instead placed the average coverage metrics for the lowest and highest covered segments within the text. This section now states: “Coding-complete sequences of all segments were de novo assembled with average coverage ranging between ~25x in the PB1 fragment and 931x in the M segment (Figure S2). Re-mapping clean reads to the putative virus genome indicated that chum salmon influenza-like virus RNA corresponded to 0.045% of the library (45,151/100,244,990 mapped reads).” [lines 189-190]
Figure 2. Wenling hagfish influenza virus is used as an outgroup for their phylogenies. I'm curious what the level of divergence is of these novel viruses with the Isaviruses? for example how much more we have to look to reconstruct the relationships and evolution of the Orthomyxoviridae as a whole. Depending on the level of divergence among these, the authors might want to include or not a related statement.
Response: The viruses described here fall into the cluster containing influenza viruses, with Wenling Hagfish virus the most divergent. Family-wide phylogenies of orthomyxoviruses and orthomyxovirus-like viruses (for example Wille & Holmes 2019 (link below)) consistently show Isaviruses as very distantly related to the influenza-like virus grouping. The inclusion of Isaviruses as a more divergent outgroup therefore provides no benefit and would adversely impact the scaling of the phylogenies.
Wille & Holmes 2019. http://perspectivesinmedicine.cshlp.org/content/10/7/a038489/F1.expansion.html
Figure 3. Please increase the size of the scale bars and support values. They are not readable in the current version.
Response: The size of the scale bar and support values for relevant nodes have been increased.
Round 2
Reviewer 1 Report
I appreciate that the authors significantly improved the manuscript. Although I am not convinced by the authors' discussion about "co-divergence," the dispute must be based on different perspectives on evolution and phylogenetics between the authors and myself. Discussion is discussion. I do not think that would undermine the importance of the study. Above all, their findings are totally worth publishing for other researchers in this field.